# Studies on the Prediction and Extraction of Methanol and Dimethyl Carbonate by Hydroxyl Ammonium Ionic Liquids

**DOI:** 10.3390/molecules28052312

**Published:** 2023-03-02

**Authors:** Xiaokang Wang, Yuanyuan Cui, Yingying Song, Yifan Liu, Junping Zhang, Songsong Chen, Li Dong, Xiangping Zhang

**Affiliations:** 1Henan Institute of Advanced Technology, Zhengzhou University, Zhengzhou 450003, China; 2Beijing Key Laboratory of Ionic Liquids Clean Process, CAS Key Laboratory of Green Process and Engineering, State Key Laboratory of Multiphase Complex Systems, Institute of Process Engineering, Chinese Academy of Sciences, Beijing 100190, China; 3Advanced Energy Science and Technology Guangdong Laboratory, Huizhou 516003, China

**Keywords:** COSMO-RS, ionic liquids (ILs), dimethyl carbonate (DMC), methanol, extraction separation

## Abstract

The separation of dimethyl carbonate (DMC) and methanol is of great significance in industry. In this study, ionic liquids (ILs) were employed as extractants for the efficient separation of methanol from DMC. Using the COSMO-RS model, the extraction performance of ILs consisting of 22 anions and 15 cations was calculated, and the results showed that the extraction performance of ILs with hydroxylamine as the cation was much better. The extraction mechanism of these functionalized ILs was analyzed by molecular interaction and the σ-profile method. The results showed that the hydrogen bonding energy dominated the interaction force between the IL and methanol, and the molecular interaction between the IL and DMC was mainly Van der Waals force. The molecular interaction changes with the type of anion and cation, which in turn affects the extraction performance of ILs. Five hydroxyl ammonium ILs were screened and synthesized for extraction experiments to verify the reliability of the COSMO-RS model. The results showed that the order of selectivity of ILs predicted by the COSMO-RS model was consistent with the experimental results, and ethanolamine acetate ([MEA][Ac]) had the best extraction performance. After four regeneration and reuse cycles, the extraction performance of [MEA][Ac] was not notably reduced, and it is expected to have industrial applications in the separation of methanol and DMC.

## 1. Introduction

Dimethyl carbonate (DMC) is an important high value-added chemical [1]. Due to its high dielectric constant, DMC can be used as electrolyte for lithium electronic batteries [2]. DMC is considered a potential fuel additive due to its high oxygen content (53%), high octane number, high gasoline/water partition coefficient, low toxicity, and fast biodegradation [3]. Engine soot particle emissions may be decreased with the right quantity of DMC added to diesel, which will lower environmental pollution [4]. Since DMC contains the methoxyl, methyl and carbonyl groups, it is also a promising replacement for toxic dimethyl sulfate and phosgene in the synthesis of isocyanates, polyurethanes and polycarbonates [5].

Currently, phosgene [6], methanol oxidation carbonylation [7], transesterification [8], urea alcoholysis [9], and direct carbon dioxide synthesis [10] are the main methods to synthesize DMC. In these routes, the separation of DMC and methanol is a difficult problem due to the formation of azeotrope [11]. Conventional separation techniques, such as pressure-swing distillation [11], azeotropic distillation [12], extractive distillation [13], membrane separation [14] and adsorption [15], have been used to separate methanol-DMC azeotropic mixtures. However, these separation techniques have high energy consumption, which is typical of thermal-based separation techniques. Liquid-liquid extraction is a form of separation technique that is popular in the chemical industry, with excellent environmental friendliness and low energy performance. To date, some research on the extraction and separation of alcohols and esters has been reported. Yang et al. [16] evaluated the solubility and liquid-liquid equilibrium data of the ternary system of dimethyl adipate + 1,6-hexanediol + water or ethylene glycol. The results showed that water is an appropriate solvent for 1,6-hexanediol extraction. Liu et al. [17] used deep eutectic solvents consisting of choline chloride and ethylene glycol to separate methanol and DMC. The experimental results showed that the extractant can realize the separation of DMC and methanol, and the maximum partition coefficient of methanol in the two phases is 0.1516. In general, using organic solvents as extractants is not environmentally friendly and it is difficult to regenerate and recycle.

Ionic liquid (IL), characterized by low saturated vapor pressure, good stability and designability, has excellent solubility for many systems and is an ideal solvent, frequently substituting for organic solvents in numerous industrial fields, particularly the extraction process [18]. However, the high viscosity and high cost of ILs have limited their industrial development to some extent. Cai et al. [19] reported the liquid-liquid equilibrium data of the ternary system of methanol + DMC + 1-alkyl-3-methylimidazolium dialkylphosphate. However, the selectivity and partition coefficient of the IL are low. Wen et al. [20] measured the liquid-liquid equilibrium (LLE) data of the ternary system of methanol + DMC + 1-methylmidazole hydrogen sulfate ([MIM][HSO_4_]) at 298.15 K and 318.15 K. The results showed that [MIM][HSO_4_] is a potential solvent for the separation of methanol and DMC. Most of the existing studies have focused on the determination of phase equilibrium data of ternary systems, and it is urgent to develop new extractants.

By introducing ILs into the methanol-DMC binary azeotropic system, the interaction between methanol and DMC can be regulated by the design of IL anions and cations [21]. The composition of ILs is diverse, and it is unrealistic to experiment one by one. Based on quantum chemistry theory and statistical thermodynamics theory, the COSMO-RS model is suitable for predicting the thermodynamic properties of LLE systems containing ILs, which can greatly reduce the blindness of IL screening and the workload of experiments [22]. To date, there has been much work on COSMO-RS screening of ILs for extraction. Jiang et al. [23] used COSMO-RS to screen suitable ILs from imidazole and pyridine ILs to extract 1,5-pentanediamine from aqueous solution. Zhao et al. [24] successfully obtained an IL with high efficiency for extracting lithium from aqueous solution by COSMO-RS simulation and experimental validation. To obtain the best IL for the drying of chloromethane, Wang et al. [25] screened 210 ILs using the COSMO-RS model and selected [EMIM][BF_4_] as the water-removing agent. There has been no systematic study on extractant screening and extraction separation for the liquid-liquid extraction separation of a methanol-DMC azeotropic system.

In this work, 330 ILs composed of 15 cations and 22 anions were screened by using the COSMO-RS model. The selectivity (*S*), solvent loss (*SL*), and solvent solubility property (*SP*) of ILs for extracting methanol from DMC were evaluated. The effects of anion and cation species on the extraction performance of ILs were investigated by analyzing the molecular interaction and σ-profile. Subsequently, five hydroxyl ammonium ILs were selected and synthesized, and then extraction experiments were carried out.

## 2. Results and Discussion

### 2.1. COSMO-RS Prediction Results

Infinite dilution activity coefficients for methanol and DMC in different ILs were predicted by COSMO-RS. *S*, *SP* and *SL* were calculated and are illustrated in Figure 1, Figure 2 and Figure 3. The specific data are listed in Appendix A. The anions and cations involved in the prediction are listed in Table 1 and Table 2, whose changes have a considerable impact on the selectivity. When choosing ILs, the value of SL should be considered, which must be extremely small. The smaller the value of SL is, the lower the loss of DMC in the extraction is. In addition, we prefer to choose ILs with a high value of SP. The higher the value of SP is, the stronger the extraction capacity of IL for methanol is. The results show that the ILs of alcohol amine cations and carboxylic acid anions have good performance and can be used to extract and separate methanol and DMC systems. Therefore, we selected these ILs for further study.

### 2.2. Molecular Interaction Analysis

One of the most significant molecule-specific characteristics in COSMO-RS theory is the σ-profile, which is often divided into three regions: the nonpolar region (−0.0082 e/Å^2^ < σ < 0.0082 e/Å^2^), the hydrogen bond acceptor (HBA) region (s > 0.0082 e/Å^2^), and the hydrogen bond donor (HBD) region (s < −0.0082 e/Å^2^) [23]. The impacts of the cation and anion species on the effectiveness of IL extraction were investigated by σ-profiles and molecular interactions.

The σ-profiles of methanol and DMC are shown in Figure 4. It can be seen from the figure that DMC has a peak at 0.0125 e/Å^2^, representing the carbonyl group and methoxy group in the molecule, which has strong hydrogen bond receiving capabilities. Methanol has peaks at −0.016 e/Å^2^ and 0.017 e/Å^2^, respectively, because the hydroxyl group in the molecular structure has strong hydrogen bond receiving (oxygen atom) and hydrogen bond supplying (hydrogen atom) capabilities at the same time, which results in strong intermolecular interactions between methanol and DMC, making it difficult to separate them. However, the possibility of separation by extraction is increased by the variation in the σ-profile distribution between DMC and methanol in the HB donor region.

#### 2.2.1. Effect of Anions on Molecular Interaction

To further understand the influence of molecular interactions on the extraction performance of ILs, COSMOthermX was used to calculate the magnitude of three kinds of interaction forces of ILs on methanol and DMC, which are misfit, van der Waals (VDW), and hydrogen-bond (HB) interactions according to COSMO-RS theory [26]. Positive energy means repulsion, while negative energy means attraction [27].

The molecular interactions of ILs composed of [MEA]^+^ and six anions on methanol and DMC were investigated, and are shown in Figure 5 and Figure 6. From Figure 5 we can see that the total interaction between ILs and methanol was much greater than that between ILs and DMC, which is necessary for the effective separation of methanol from DMC. Figure 6a shows that the molecular interaction between methanol and ILs is mainly hydrogen bonding. With the enhancement of acid radical ion acidity, the hydrogen bonding force gradually increases, resulting in a slightly increasing *SP* value. Figure 6b shows that the dominant molecular interaction between DMC and ILs is the VDW interaction. With the enhancement of acid radical ion acidity, the VDW interaction gradually increases, which leads to a significant increase in the *SL* value, resulting in a decrease in the *S* value. The *S* value follows the order: [MEA][NO_3_] > [MEA][HSO_4_] > [MEA][HCO_3_] > [MEA][Frc] > [MEA][Ac] > [MEA][Prp].

In addition, the σ-profiles of the six anions were also analyzed and are shown in Figure 7. The six anions have peaks mainly in the HBA region, and the peak height in the HBA region is very high, which indicates that these anions are more capable of accepting hydrogen bonds. As mentioned above, methanol has a high capacity for supplying hydrogen bonds, so methanol can be extracted by forming a strong hydrogen bond with the IL. Meanwhile, the entrainment of DMC during extraction can be effectively reduced due to the weak ability of DMC to provide hydrogen bonds.

#### 2.2.2. Effect of Cations on Molecular Interaction

The effect of cations on the extraction ability of ILs is less significant than that of anions. Fixing the anion to [Ac]^−^, the effects of the number of hydroxyethyl groups on the cation on the interactions between ILs and methanol and between ILs and DMC were investigated.

It can be seen from Figure 8 that the total interaction between ILs and methanol is much stronger than that between ILs and DMC. Figure 9a shows that there is a weak increase in the magnitude of the hydrogen bonding force between the IL and methanol, as the number of hydroxyethyl groups rises, which increases the value of *SP*. Figure 9b shows that the increase in the number of hydroxyethyl groups also caused an increase in the magnitude of van der Waals and hydrogen bonding forces between DMC and IL, which caused a significant increase in the value of *SL*. Since *SL* rises faster than *SP*, the *S* value keeps decreasing. The *S* value follows the order: [MEA][Ac] > [DEA][Ac] > [TEA][Ac].

Similarly, the σ-profiles of the three cations were analyzed and are shown in Figure 10. The result shows that as the number of hydroxyethyl groups rises, the peak of the IL extends further in the HB donor region and higher in the HB acceptor region. This implies that the IL has a greater capacity for donating and accepting hydrogen bonds, which significantly improves its affinity for both DMC and methanol. Inevitably, more DMC is entrained despite the increased IL extraction efficiency for methanol.

### 2.3. Extraction Experiment

Due to the instability of [MEA][HCO_3_] and the difficulty of synthesizing [MEA][NO_3_] and [MEA][HSO_4_], five ILs: [MEA][Frc], [MEA][Ac], [MEA][Prp], [DEA][Ac] and [TEA][Ac] were synthesized. All ILs are characterized by ^1^H NMR spectra, which are listed in Appendix A. This confirmed the structures of the ILs and that there was no impurity in the ILs. The decomposing temperatures of the synthesized ILs were more than 140 °C, which is relatively stable under experimental conditions [28].

Five ILs were used to carry out extraction experiments. Through the experiments, the separation performance of typical ILs was examined to confirm the validity of COSMO-RS. At 293.15 K, the two-phase composition after the experiment was measured, and the experimental results are listed in Table 1 and Table 2.

Among the ILs, [MEA][Frc], [MEA][Ac] and [MEA][Prp] were employed to investigate the impact of various anions on the extraction effect. The results showed that the three ILs had high extraction efficiency and selectivity, and the order of *S* was [MEA][Frc] > [MEA][Ac] > [MEA][Prp]. As the anion changes from [Frc]^−^ to [Ac]^−^ and then to [Prp]^-^, *S* is significantly reduced. This is because as the anion’s alkyl chain lengthens, the volume of the entire IL expands and its polarity decreases [29]. As a result, the van der Waals force between DMC and the IL develops, which lowers *S*.

Similarly, to investigate the effect of cation types on the extraction performance of ILs, we selected three ILs [MEA][Ac], [DEA][Ac] and [TEA][Ac] for extraction experiments, and the results are listed in Table 2. It can be seen from the table that the order of *S* is [MEA][Ac] > [DEA][Ac] > [TEA][Ac]. Compared with anions, the variety of cation types does not bring a noteworthy change to the selectivity of the IL. The reasons may include the following aspects. On the one hand, the increase in hydroxyl groups greatly strengthens the hydrogen bonding between the IL and methanol. On the other hand, when the number of hydroxyethyl groups attached to the N atom rises, the IL polarity declines, increasing the van der Waals force between the IL and DMC. Given these two factors, it is not unexpected that the value of *S* only marginally decreases.

In addition, it can be found in experiments that with the increase in the initial concentration of methanol, the S value of ILs decreases, but E slightly increases. Most ILs show a high extraction efficiency of more than 80%.

### 2.4. Recyclability

In the extraction experiment, the recycling of the extractant is also an important aspect. It is necessary to regenerate and recycle ILs from the mixture of methanol, DMC and ILs. After extraction, IL was regenerated from the IL phase by vacuum evaporation in 343.15 K for 5 h. Taking [MEA][Ac] as an example, using recycled [MEA][Ac] each time, we carried out extraction experiments under the same conditions four times. As shown in Figure 11, after four cycles, the extraction efficiency of [MEA][Ac] decreased slightly from 86.1% to 84.2%, proving the stability and efficiency of [MEA][Ac]. The fresh and recycled [MEA][Ac] were then analyzed by ^1^H NMR, and the results are shown in Appendix A. The results showed that the ^1^H NMR spectra of fresh and recycled ILs did not change, indicating that the structure of recycled ILs did not change.

## 3. Materials and Methods

### 3.1. Computational Methods

The COSMOthermX program was used to perform all COSMO-RS calculations at the BVP86/TZVP quantum chemical level and related parameterization (BP TZVP C30 1401). The .cosmo files of methanol, DMC, and some anions and cations are obtained directly from the database. For ILs that do not exist in the database, Turbomole was used to generate their corresponding .cosmo files. The molecular model was computed using Turbomole software at the RI-DFT level of theory and the def-TZVP basis set. Because the parameterization divergence between TZVP and TZVPD-FINE is very similar, the TZVP operation is much faster than that of TZVPD-FINE [30]. In this work, 330 ILs composed of 15 cations and 22 anions were screened. The anions and cations involved in the prediction are listed in Table 3 and Table 4.

### 3.2. Extraction Performance Index of ILs Using the COSMO-RS Model

The performance of methanol extraction from DMC by IL was predicted by the COSMO-RS model. Three important parameters, *S*, *SP*, and *SL*, were determined by
(1)$$S=\frac{\gamma_{\mathrm{DMC}}^{\infty }}{\gamma_{\mathrm{methanol}}^{\infty }}$$
(2)$$\mathrm{SP}=\frac{1}{\gamma_{\mathrm{methanol}}^{\infty }}$$
(3)$$\mathrm{SL}=\frac{1}{\gamma_{\mathrm{DMC}}^{\infty }}$$
where $\gamma_{\mathrm{DMC}}^{\infty }$ and $\gamma_{\mathrm{Methanol}}^{\infty }$ represent the infinite dilution activity coefficients of DMC and methanol in ILs, respectively. *S* stands for selectivity, which reflects the ability of ILs to extract methanol from DMC. The higher the value of *S*, the better the extraction separation effect. When *S* = 1, which means $\gamma_{\mathrm{DMC}}^{\infty }$ = $\gamma_{\mathrm{Methanol}}^{\infty }$, IL cannot be used for extraction. *SP* stands for the solvent solubility property, and the higher the value is, the better the solubility of the IL to methanol is. *SL* stands for solvent loss, which reflects the loss of DMC after extraction. The higher the value is, the greater the loss.

### 3.3. Materials and Reagents

The specific compounds utilized for the IL synthesis and extraction studies are shown in Table 5. All chemicals were purchased from Aladdin, and they were not purified further prior to use.

### 3.4. Synthesis of ILs

The synthesis of the ILs used in this study followed the acid-base neutralization reaction and was prepared by a one-step synthesis method [31]. Taking [MEA][AC] as an example, 100 mL of ethanol was used to dissolve 0.5 mol of 2-aminoethanol, which was then added into a 500 mL flask as a liquid mixture. As the reaction was intense and exothermic, the flask was placed in an ice water bath at 278.15 K under vigorous stirring with a magnetic stirrer. Next, 100 mL of mixed ethanol and 0.5 mol of acetic acid were slowly added to the flask within 90 min. The reaction persisted for 20 h. Following the reaction, the solvent was eliminated by vacuum evaporation, and the unreacted reactants were eliminated using repeated acetone washing. After being dried under a vacuum for 48 h at 333.15 K, a clear and colorless product was produced. It is worth mentioning that in the synthesis of these ILs, the synthesis process requires adequate cooling and strict control of the addition rate, as the reaction between formic acid and ethanolamine is the most exothermic. Otherwise, under high heat, dehydration of the salt may occur, resulting in the corresponding formamide, as in the case of nylon salt (salt of diamine and dicarboxylate) [32]. ^1^H NMR spectra were used to confirm the structure of the ILs, and there was no impurity.

### 3.5. Extraction Experiments

The specific experimental methods are described below. Prepare DMC solution with 10%, 20%, and 30% mole fractions of methanol. Add 20 g solution to a 50 mL jacketed extraction bottle, and then add the same mass of IL. Seal the jacketed glass bottle to prevent the solution from volatilizing, and circulate water outside to keep the temperature of the system stable at 293.15 K. The solution was magnetically stirred at 500 rpm for 2 h and then remained still for 10 h to guarantee balance. Finally, the lower sampling port was unscrewed to take the lower sample, and the upper sample was removed from the upper layer with a straw, and the samples were then analyzed.

### 3.6. Sample Analysis

Gas chromatography (Agilent 8890A) outfitted with a DB-624 capillary column and a flame ionization detector (FID), along with the internal standard method, was used to measure the concentrations of methanol and DMC in two phases. ILs are nonvolatile, so the composition of IL in the sample was determined by the weight loss method. The volatile organic compounds were removed using a vacuum drying oven, and the weight of each sample was precisely determined both before and after evaporation. The evaporated samples’ GC analysis showed no residues of organic materials [20].

### 3.7. Evaluation Index of Extraction Performance in Experiments

The distribution coefficient (*D*), selectivity (*S*), and extraction efficiency (*E*) are used to evaluate the extraction performance of ILs, and they are calculated as follows:(4)$$D=\frac{x_{\mathrm{methanol}}^{E}}{x_{\mathrm{methanol}}^{R}}$$
(5)$$S=\frac{x_{\mathrm{methanol}}^{E}}{x_{\mathrm{methanol}}^{R}}\times \frac{x_{\mathrm{DMC}}^{R}}{x_{\mathrm{DMC}}^{E}}$$
(6)$$E=\frac{n_{\mathrm{methanol}}^{E}}{\;n_{\mathrm{methanol}}^{E}{+n}_{\mathrm{methanol}}^{R}}$$
where $x_{\mathrm{methanol}}$ and $x_{\mathrm{DMC}}$ represent the mole fractions of methanol and DMC, $n_{\mathrm{methanol}}$ represents the molar amount of methanol, and superscripts R and E represent the raffinate phase and extraction phase, respectively.

## 4. Conclusions

In this work, 330 ILs composed of 15 cations and 22 anions were screened by the COSMO-RS model to find ILs with good extraction performance and realize the effective separation of DMC and methanol. It can be seen from the results that the type of anions have a greater influence on the extraction performance of ILs than that of cations, and carboxylic acid anions deliver the most optimal performance when paired with cations such as [MEA]^+^, [DEA]^+^ or [TEA]^+^. Through the analysis of molecular interactions, it can be concluded that the HB interaction is dominant between methanol and ILs, while VDW is the main force between DMC and ILs. Five ILs are synthesized for extraction experiments to verify the reliability of the COSMO-RS model. The order of IL selectivity obtained from the experiment is consistent with that predicted by the COSMO-RS model. All selected ILs have high selectivity and distribution coefficients. Among these ILs, [MEA][Ac] has the highest extraction efficiency, which can reach 86.81%. In addition, [MEA][Ac] has the advantages of easy regeneration and recyclability, and is expected to be applied in the separation of methanol and DMC in industrial production.

## Figures and Tables

**Figure 1 molecules-28-02312-f001:**
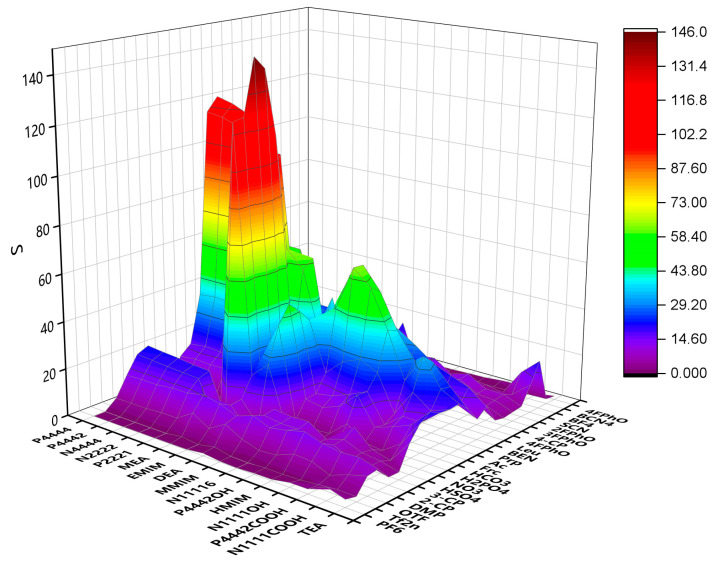
S of ILs predicted by COSMO-RS.

**Figure 2 molecules-28-02312-f002:**
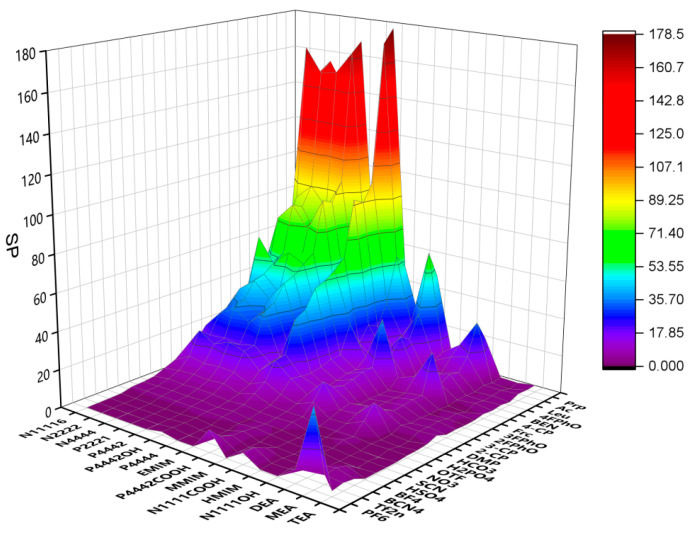
SP of ILs predicted by COSMO-RS.

**Figure 3 molecules-28-02312-f003:**
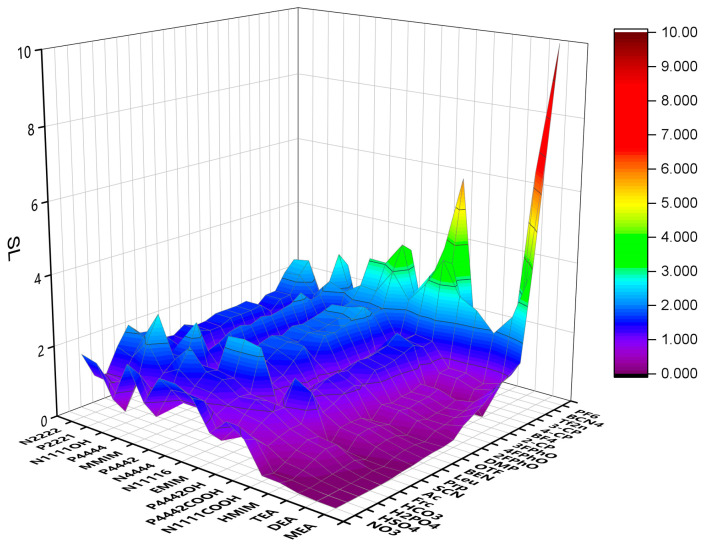
SL of ILs predicted by COSMO-RS.

**Figure 4 molecules-28-02312-f004:**
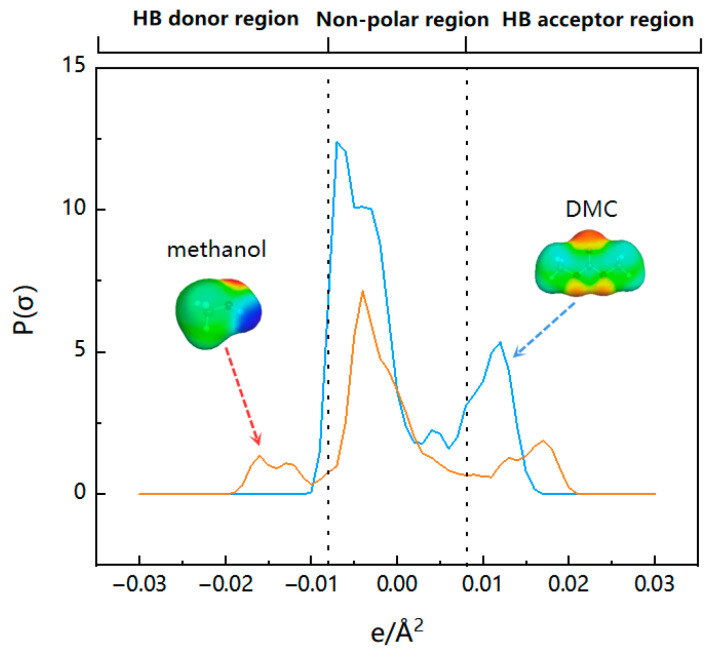
The σ-profiles of methanol and DMC.

**Figure 5 molecules-28-02312-f005:**
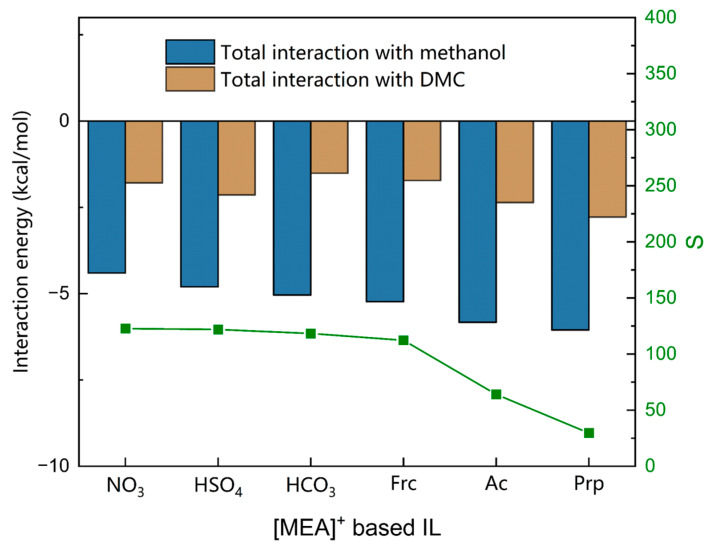
Total interaction between solutes and [MEA]^+^ based ILs calculated by COSMO-RS.

**Figure 6 molecules-28-02312-f006:**
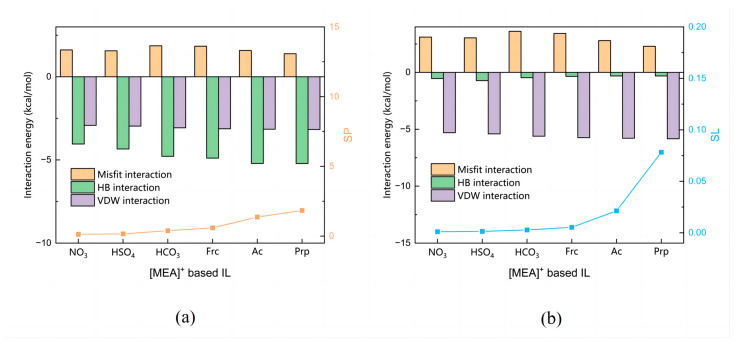
Molecular interaction calculated by COSMO-RS. (**a**) Molecular interaction between methanol and [MEA]^+^ based ILs. (**b**) Molecular interaction between DMC and [MEA]^+^ based ILs.

**Figure 7 molecules-28-02312-f007:**
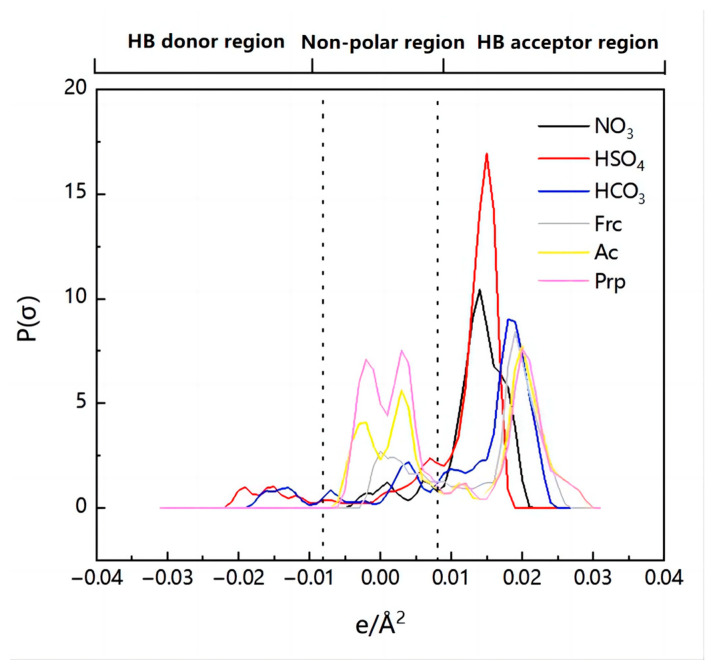
The σ-profiles of six different anions.

**Figure 8 molecules-28-02312-f008:**
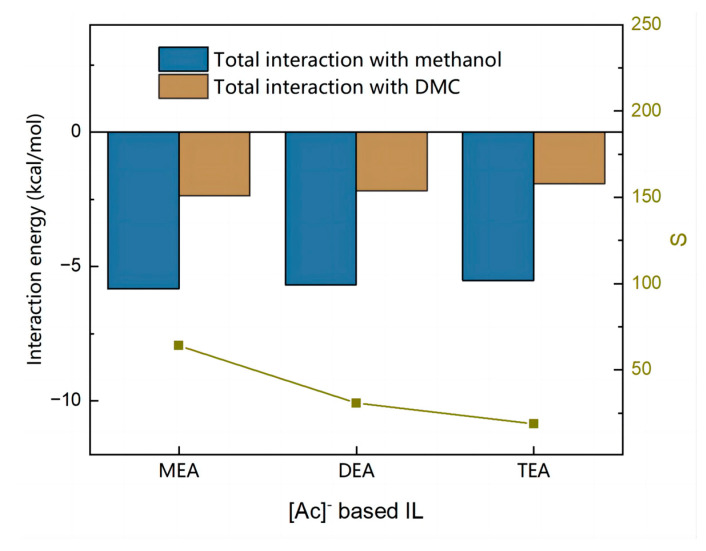
Total interaction between solutes and [Ac]^−^ based ILs calculated by COSMO-RS.

**Figure 9 molecules-28-02312-f009:**
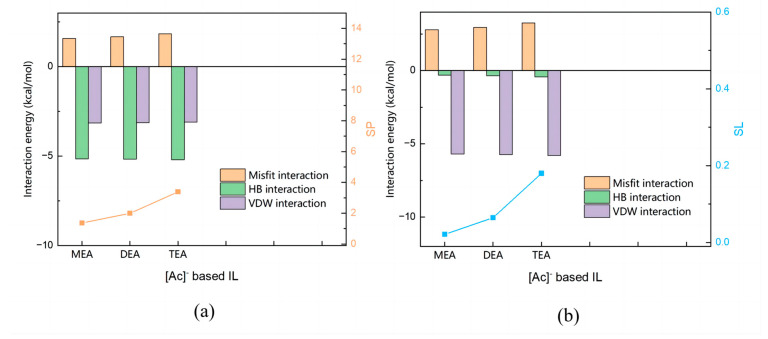
Molecular interaction calculated by COSMO-RS. (**a**) Molecular interaction between methanol and [Ac]^−^ based ILs. (**b**) Molecular interaction between DMC and [Ac]^−^ based ILs.

**Figure 10 molecules-28-02312-f010:**
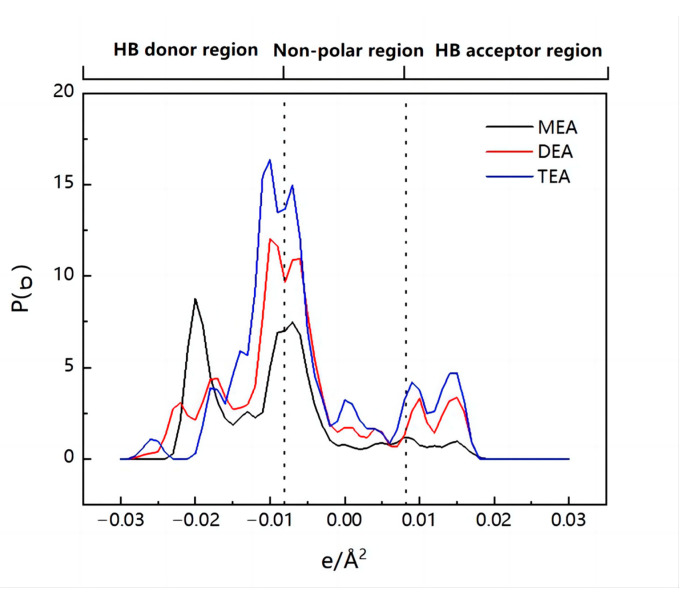
The σ-profiles of three different cations.

**Figure 11 molecules-28-02312-f011:**
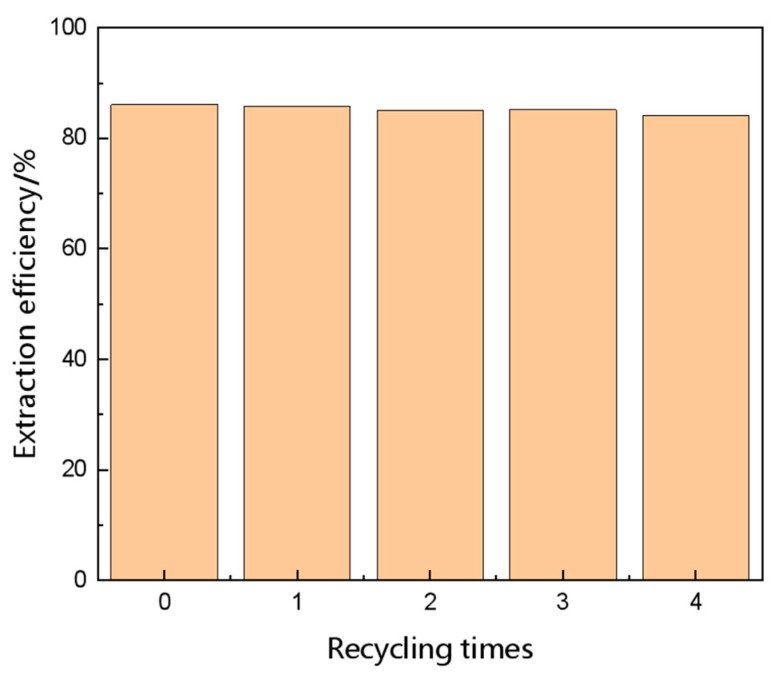
Extraction performance of recycled IL.

**Table 1 molecules-28-02312-t001:** Liquid-liquid experiment results for ILs with different types of anions at 293.15 K.

Extractant	Initial Concentration of Methanol	DMC Phase (%)			IL Phase (%)			D	S	E
		x_MeOH_	x_DMC_	x_IL_	x_MeOH_	x_DMC_	x_IL_			
[MEA][Frc]	10%	2.54	97.44	0.02	9.46	4.25	86.29	3.72	85.27	79.20%
	20%	5.50	94.46	0.03	17.72	4.59	77.69	3.22	66.23	79.25%
	30%	8.65	91.30	0.05	24.50	4.70	70.80	2.83	54.97	79.29%
[MEA][Ac]	10%	2.13	97.85	0.02	9.98	14.69	75.33	4.70	31.29	83.31%
	20%	4.10	95.86	0.04	19.08	15.32	65.60	4.65	29.10	85.84%
	30%	6.86	93.09	0.05	27.60	16.34	56.06	4.02	22.92	86.81%
[MEA[Prp]	10%	2.23	97.52	0.25	10.54	15.40	74.07	4.73	29.93	83.24%
	20%	4.73	94.99	0.27	18.95	16.90	64.15	4.00	22.50	83.76%
	30%	7.24	92.42	0.35	28.28	17.99	53.74	3.91	20.07	86.93%

**Table 2 molecules-28-02312-t002:** Liquid-liquid experiment results for ILs with different types of cations at 293.15 K.

Extractant	Initial Concentration of Methanol	DMC Phase (%)			IL Phase (%)			D	S	E
		x_MeOH_	x_DMC_	x_IL_	x_MeOH_	x_DMC_	x_IL_			
[MEA][Ac]	10%	2.13	97.85	0.02	9.98	14.69	75.33	4.70	31.29	83.31%
	20%	4.10	95.86	0.04	19.08	15.32	65.60	4.65	29.10	85.84%
	30%	6.86	93.09	0.05	27.60	16.34	56.06	4.02	22.92	86.81%
[DEA][Ac]	10%	2.66	97.05	0.29	12.44	15.75	71.81	4.68	28.84	80.77%
	20%	4.70	94.95	0.35	19.70	20.58	59.72	4.20	19.35	82.97%
	30%	7.99	91.62	0.39	31.14	21.12	47.74	3.90	16.91	86.13%
[TEA][Ac]	10%	2.50	97.04	0.46	13.25	22.53	64.22	5.29	22.80	80.04%
	20%	5.06	94.46	0.48	23.61	23.09	53.30	4.67	19.10	81.78%
	30%	8.31	91.16	0.53	33.25	23.70	43.04	4.00	15.39	83.99%

**Table 3 molecules-28-02312-t003:** The names of the cations.

Number	Names	Abbreviations
1	2-Hydroxyethylammonium	[MEA]^+^
2	Bis(2-hydroxyethyl)ammonium	[DEA]^+^
3	Tris(2-hydroxyethyl)ammonium	[TEA]^+^
4	1-Ethyl-3-methylimidazolium	[EMIM]^+^
5	3-Methylimidazolium	[HMIM]^+^
6	1-Methyl-3-methylimidazolium	[MMIM]^+^
7	Hexadecyltrimethylammonium	[N_11116_]^+^
8	Hydroxyltrimethylammonium	[N_1111OH_]^+^
9	Carboxyltrimethylammonium	[N_1111COOH_]^+^
10	Tetraethylammonium	[N_2222_]^+^
11	Tetrabutylammonium	[N_4444_]^+^
12	Methyltriethylphosphorus	[P_2221_]^+^
13	Ethyltributylphosphorus	[P_4442_]^+^
14	Hydroxyethyltributylphosphorus	[P_4442OH_]^+^
15	Tetrabutylphosphorus	[P_4444_]^+^

**Table 4 molecules-28-02312-t004:** The names of the anions.

Number	Names	Abbreviations
1	Hexafluorophosphate	[PF_6_]^−^
2	Bis(trifluoromethylsulfonyl)imide	[Tf_2_N]^−^
3	Tetracyanoboric acid	[BCN_4_]^−^
4	Tetrafluoroborate	[BF_4_]^−^
5	Hydrogen sulfate	[HSO_4_]^−^
6	Thiocyanate thiocyanide	[SCN]^−^
7	Nitrate	[NO_3_]^−^
8	Trifluoromethanesulfonate	[OTf]^−^
9	Dihydrogen phosphate	[H_2_PO_4_]^−^
10	Hydrocarbonate	[HCO_3_]^−^
11	Dimethylphosphate	[DMP]^−^
12	2-Chlorophenol	[2-CP]^−^
13	3-Chlorophenol	[3-CP]^−^
14	Difluorophosphate	[2FPhO]^−^
15	Trifluorophosphate	[3FPhO]^−^
16	Formate	[Frc]^−^
17	4-Chlorophenol	[4-CP]^−^
18	Benzoate	[BEN]^−^
19	Tetrafluorophosphate	[4FPhO]^−^
20	Leucinate	[Leu]^−^
21	Acetate	[Ac]^−^
22	Propionate	[Prp]^−^

**Table 5 molecules-28-02312-t005:** Materials.

Names	CAS	Purity	Supplier
Dimethyl carbonate	616-38-6	≥99%	Aladdin
Methanol	67-56-1	≥99%	Aladdin
Ethanolamine	141-43-5	≥99%	Aladdin
Diethanolamine	111-42-2	≥99%	Aladdin
Triethanolamine	102-71-6	≥99%	Aladdin
Formic acid	64-18-6	≥99%	Aladdin
Acetic acid	64-19-7	≥99%	Aladdin
Propanoic acid	79-09-4	≥99%	Aladdin

## Data Availability

Not applicable.

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
