# Peer review of "Studies on the Prediction and Extraction of Methanol and Dimethyl Carbonate by Hydroxyl Ammonium Ionic Liquids"

_molecules, 2023, doi:10.3390/molecules28052312_

Round 1

Reviewer 1 Report

The authors studied the prediction and extraction experiment of methanol 2 and dimethyl carbonate by hydroxyl ammonium ionic liquids. The results of the work are interesting. However, some aspects need to be clarified:

1.      In the abstract, indicate the value of the consistency of the prediction with the COSMO program with respect to the experimental results.

2.      In the abstract in lines 15 and 16 it must be indicated that the "hydroxylammonium " corresponds to the cation.

3.      In page 2, lines 52-56. Why are DES not appropriate for this type of extraction? The authors point out that the partition coefficient is low. However, there are different types of DES, which perhaps improve the partition coefficients. Please explain better.

4.      In the paragraph of the page 2 lines 58-60, only the advantages of the IL are mentioned. However, they are known to have many drawbacks. please mention them.

5.      On page 2 lines 84-89 it is not clear if in the work if I use COSO-RS it was used to select the most appropriate ILs, to then select a few to carry out the experimental work. Or if the work is completely done in silico.

6.      On page 3 lines 97-98 the sentence is not understood. Does the SP value have to be high or low? Or do they have to be located at the extremes?

7.      Improve the resolution of figures 1, 2 and 3, since the initials of the compounds that go on the axes are not distinguished.

8.      Figures 6 and 7 should be left as one figure and put 6A and 6B. The same for figures 10 and 11.

Reviewer 2 Report

The English form is borderline incomprehensible. The paper must undergo extensive English revision before any further handling.

This referee would like to see the NMR spectra on the prepared ILs, including peak integral analysis. It is known that Triethylammonium Acetate is not forming as an ionic liquid, separating in two liquid phases. The bottom phase was mistaken as the pure ionic liquid (see DOI 10.1016/j.molliq.2020.115069). The authors must provide evidence that their ILs are actually 50:50 acid:base ILs.

Reviewer 3 Report

The article reports modeling and experimental results showing promising extraction of DMC out of methanol with functionalized ILs with hydroxyl ammonium groups. The driving force is the difference between IL/methanol hydrogen bonding and IL/DMC VDW. [MEA][Ac] has the best extraction performance. I would recommend publication after some revisions:

1. What are the measured or reference volatility values for the synthesized ILs?

2. The NMR results need attentions. The peak integrals for all spectrums are not correlated to the design structures. Based on the current results, the synthesized IL molecules are different from the proposed molecules. 

Round 2

Reviewer 1 Report

This new version incorporates all the observations. Therefore I recommend its publication.